# ADVERSARIAL INTERPOLATION TRAINING
## A SIMPLE APPROACH FOR IMPROVING MODEL ROBUSTNESS

## ABSTRACT

We propose a simple approach for adversarial training. The proposed approach utilizes an adversarial interpolation scheme for generating adversarial images and accompanying adversarial labels, which are then used in place of the original data for model training. The proposed approach is intuitive to understand, simple to implement and achieves state-of-the-art performance. We evaluate the proposed approach on a number of datasets including CIFAR10, CIFAR100 and SVHN. Extensive empirical results compared with several state-of-the-art methods against different attacks verify the effectiveness of the proposed approach.

## 1 INTRODUCTION

Deep learning-based techniques have achieved outstanding performance in many tasks such as image classification (Krizhevsky et al., 2012), speech recognition (Hinton et al., 2012) and video game playing (Mnih et al., 2015). Despite of these encouraging progress, it has been shown that these models could be easily attacked by adversarial examples (Szegedy et al., 2014; Biggio et al., 2013; Carlini & Wagner, 2018; Lin et al., 2017). The ubiquitous of adversarial examples across many tasks (Szegedy et al., 2014; Lin et al., 2017; Eykholt et al., 2018; Carlini & Wagner, 2018) and the fact that they are transferable between different models (Tramèr et al., 2017; Charles et al., 2019; Moosavi-Dezfooli et al., 2017) raise great concerns on security of such models and hinder their actual deployment in real world applications. Researchers have been actively working on understanding the cause of adversarial examples (Szegedy et al., 2014; Goodfellow et al., 2015) and approaches for improving model robustness against them (Madry et al., 2018; Tramèr et al., 2018; Liao et al., 2018; Yan et al., 2018). A number of theories have been developed for explaining the existence of adversarial examples. Szegedy et al. (2014) explain that adversarial examples are possible because the image space is densely filled with low probability adversarial pockets. Goodfellow et al. (2015) argue that the adversarial examples are caused by the linear nature of deep networks. Tanay & Griffin (2016) provide the perspective that adversarial examples exist because the class boundary extends beyond the data sub-manifold and can be lying close to it in some cases. It has been further shown in some recent work that adversarial examples can be decomposed into categories with different causes including off-manifold ones (Stutz et al., 2019; Jacobsen et al., 2019) and those due to natural test error (Stutz et al., 2019; Jacobsen et al., 2019; Ford et al., 2019).

Recently, Ilyas et al. (2019) provide a perspective that adversarial vulnerability is caused by non-robust features. The reasoning is that there are abundant of useful correlations that exist in natural data, thus it is natural to expect the models could learn to exploit any of them if no preference is given (*c.f.* Table 7). However, models relying on superficial statistics (non-robust features) can be brittle and generalize poorly, thus suffering from adversarial attacks (Ilyas et al., 2019). The natural idea is therefore using only robust features for learning. However robust features are not easy to construct directly (Madaan & Hwang, 2019). In contrast, non-robust features are much easier to construct, using the standard adversarial example generation procedure (Szegedy et al., 2014; Goodfellow et al., 2015; Madry et al., 2018). Therefore, we can leverage non-robust features instead for robust learning. This is typically achieved by constructing a robustified dataset where the new (perturbed) images are constructed by adding non-robust features to the clean images, and then performing model training using the perturbed images in place of the original ones. This procedure is essentially the standard *adversarial training* approach (Goodfellow et al., 2015; Madry et al., 2018), which has been shown to be effective for defending against adversarial attacks (Athalye et al., 2018).

In this paper, we present a simple approach for improving model robustness under the framework of adversarial training. We craft adversarial examples by performing a so-called *adversarial interpolation* operation (Section 3) between samples, and we expect the model should perform robustly against adversarial interpolation-induced perturbations within a constrained neighborhood.

The contribution of this work can be summarized as follows:

- we propose an adversarial interpolation approach for generating adversarial samples, which is simple, intuitive and effective;
- we analyze the proposed approach and its connections with previous methods;
- we leverage adversarial interpolation for adversarial training and verify its effectiveness in improving model robustness compared with a number of strong baselines across several datasets.

## 2 PRELIMINARIES

### 2.1 ADVERSARIAL ATTACK AND DEFENSE

Adversarial examples have been investigated in the seminal work of Biggio et al. (2013); Szegedy et al. (2014) and have attracted increasing attention recently (Biggio et al., 2013; Goodfellow et al., 2015; Tramèr et al., 2018; Madry et al., 2018; Athalye et al., 2018; Biggio & Roli, 2017; Wang et al., 2019). Szegedy et al. (2014) investigated the vulnerabilities of CNNs to adversarial examples and proposed an L-BFGS-based approach for attack generation. Goodfellow et al. (2015) developed a fast gradient sign method (FGSM) for generating adversarial images. After that, many types of attacks have been developed in the past few years (Moosavi-Dezfooli et al., 2016; Carlini & Wagner, 2017; Su et al., 2017; Xiao et al., 2018; Brown et al., 2017; Brendel et al., 2018). On the defense side, many researchers are actively working on improving model robustness against adversarial attacks Meng & Chen (2017); Xie et al. (2018); Metzen et al. (2017); Liu et al. (2018); Song et al. (2017); Liao et al. (2018); Samangouei et al. (2018); Prakash et al. (2018); Wang & Zhang (2019). Recently, Athalye et al. (2018) pointed out a phenomenon called gradient masking and showed that many existing defence methods did not actually improve model robustness thus giving a false sense of robustness because of it. According to Athalye et al. (2018) , adversarial training (Goodfellow et al., 2015; Madry et al., 2018) is one of the effective defense method against adversarial attacks. It improves model robustness by solving a minimax problem (Goodfellow et al., 2015; Madry et al., 2018) as $\min_\theta \left[ \max_{\tilde{x} \in B(x,\epsilon)} \mathcal{L}(\tilde{x}, y; \theta) \right]$, where the inner maximization is responsible for providing the most challenging example for the current model while the outer optimization loop improves the model performance on these adversarial examples (Madry et al., 2018). The inner optimization can be solved approximately by a one-step approach such as FGSM (Goodfellow et al., 2015), or a multi-step projected gradient descent (PGD) method (Madry et al., 2018).

### 2.2 INCORPORATING REGULARIZATION IN MODEL TRAINING

Regularization is important for deep learning models with a large number of parameters, as there are multiple possible solutions that perform equally well on clean data (Garipov et al., 2018; Draxler et al., 2018; Athiwaratkun et al., 2019), the model by default has no particular preference over any of them if no preferences are given. In the case of model robustness, we need to have a way to convey our preference on robustness to the learning process. This can be viewed as a form of regularization. Conventional regularization techniques include weight decay or other forms of constraints on component function (Arjovsky et al., 2017). Data augmentation is another way to regularization by manipulating the input. The idea is that an nuisance factor can be de-emphasized by generating training data that incorporates variations of the factor, leading to invariance *w.r.t.* this factor. Following this idea, conventional data augmentation transforms the training images using *label preserving* transformations (Krizhevsky et al., 2012). Zhang et al. (2018) proposed a mixup approach which *jointly transforms images and labels*. It generates training examples (both image and label) by linear interpolation between pairs of natural examples, thus introducing a linear inductive bias in the vicinity of training samples. Similar to standard data augmentation, adversarial training incorporates implicit regularization by perturbing the training data using non-robust features of *other* classes thus reducing the sensitivity of the model *w.r.t.* non-robust features (*c.f.* Table 7). It has been shown that adversarial training is equivalent to regularized training with a data dependent regularization term (Ororbia et al., 2017). Since adversarial training incorporates preference to robustness against perceptual insensitive perturbations, the models trained this way have been shown to have better interpretability (Engstrom et al., 2019; Stutz et al., 2019).

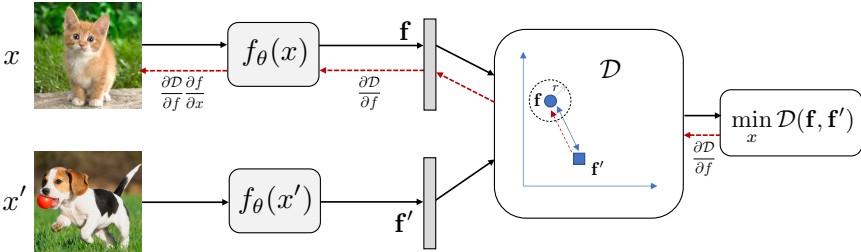

Figure 1: **Adversarial Interpolation for adversarial image generation**. It takes a clean image $x$ to be perturbed, and another distinct image $x'$, performs feature extraction using the feature extractor $f_\theta(\cdot)$ and then measures the distance in the feature space using a distance measure $\mathcal{D}$. The perturbed image is generated by minimizing the feature space distance with respect to $x'$.

## 3 ADVERSARIAL INTERPOLATION TRAINING

We present the adversarial interpolation training method in this section. The intuition is that the semantic meaning of the image is unchanged under mild perturbations, thus the network should be robust against them as well. On the one hand, it is well-known that adding random perturbations to the input is ineffective for improving model robustness. On the other hand, adversarial training using classification loss generated perturbations suffers from label leaking (Kurakin et al., 2017). These motivate the development of the proposed method as detailed in the sequel.

### 3.1 ADVERSARIAL INTERPOLATION

We first review several existing types of perturbations in the sequel and then introduce our proposed approach. We will use $\tilde{x}$ to denote the perturbed version of $x$.

**Definition 1** *Random Perturbation. Given a sample $x$, the randomly perturbed sample $\tilde{x}$ is generated by sampling from the neighborhood ball $B(x, \epsilon)$ centered at $x$ with radius $\epsilon$ as $\tilde{x} \sim B(x, \epsilon)$.*

It is clear that the random perturbations are data-agnostic and isotropic in each direction. It has been shown that this types of perturbation is equivalent to Tikhonov regularization (Bishop, 1995) and is less effective in improving model robustness due to its data and model agnostic nature. Note that we have omitted the pixel value range constraint to avoid notation clutter. Instead of randomly sampling, (discriminative) adversarial perturbations are generated with the guidance of a classification loss.

**Definition 2** *Discriminative Perturbation. Given a sample $x$ and its label $y$, we define the discriminative perturbation as:* $\quad \tilde{x}^* \triangleq \arg\min_{\tilde{x} \in B(x, \epsilon)} -\mathrm{CE}(\tilde{x}, y),$

where $\mathrm{CE}(\cdot)$ denotes the network taking $\tilde{x}$ as input and computes the cross-entropy (CE) loss between the predicted class probabilities and the input label $y$. Discriminative perturbation is data-dependent and model/task-aware (*i.e.* white-box), representing a form of strong attacks for a given model. Naturally, it has been used in adversarial training for online generation of training data that is adversarial to the current model. It is known that the discriminative perturbations are highly correlated with the decision boundary. The high correlation between the perturbation and class label poses the training to the danger of label leaking (Kurakin et al., 2017).

**Definition 3** *Adversarial Interpolation. Given two images $x, x'$, we define the adversarial interpolation of $x$ (towards $x'$) as the solution to the following optimization problem*

$$\tilde{x}^* \triangleq \texttt{adv\_interp}(x, x') = \arg\min_{\tilde{x} \in B(x, \epsilon)} \mathcal{D}(\tilde{x}, x'), \tag{1}$$

where $x'$ is another instance that is different from $x$. In practice, $x'$ can be set as another instance from the current batch, thus introducing a negligible computational overhead. $\mathcal{D}$ is a distance measure in a feature space. As minimization of $\mathcal{D}(\tilde{x}, x')$ *w.r.t.* $\tilde{x}$ has the effect of "interpolating" $x$ towards $x'$, we therefore term the proposed approach for generating adversarial attacks as *adversarial interpolation*. The adversarial interpolation procedure is illustrated in Figure 1. It leverages the currently learned model for feature extraction and distance computation. It then generates the adverse examples by back-propagating the distance induced gradient (feature space difference) through the network back to pixels. It is more effective than random perturbation as it leverages structures between data points. And it is potentially more flexible than discriminative perturbations as adversarial interpolation is used for generating perturbations which is untied from the classification loss. For a particular sample, when viewed collectively over time, adversarial interpolation essentially constructs a randomly connected graph (Hein et al., 2007) and performs interpolation along different edge each time randomly. More on understanding of the proposed approach is in Section 3.3.

---

**Algorithm 1** Adversarial Interpolation Training

---

**Input:** training epochs $K$, learning rate $\alpha$, budget $\epsilon$, $\epsilon_y$, attack iterations $L$, number of classes $C$
**for** $k = 1$ **to** $K$ **do**
    **for** random batch $\{x_i, y_i\}_{i=1}^n \sim$ dataset **do**
      $\tilde{x}_i \sim B(x_i, \epsilon)$, $\tilde{y}_i = y_i$, $x'_i = x_{n-i+1}$, $\bar{y}'_i = \frac{1}{C-1}(1 - y_{n-i+1})$
      $\mathcal{P}(\tilde{x}_i) = \min_{z_i \in \mathcal{S}} \|z_i - \tilde{x}_i\|$, $\mathcal{S} = B(x_i, \epsilon) \cap [0, 255]^{\#\text{pixel}}$
      **adversarial interpolation**:
      **for** $l = 1$ **to** $L$ **do**
        $\cdot$ $\tilde{x}_i \leftarrow \mathcal{P}\big(\tilde{x}_i - \epsilon \cdot \text{sign}\big(\nabla_{\tilde{x}_i} \mathcal{D}(\tilde{x}_i, x'_i)\big)\big)$         ▷ generating adversarial images
        $\cdot$ $\tilde{y}_i \leftarrow \tilde{y}_i - \epsilon_y \cdot (\tilde{y}_i - \bar{y}'_i)$         ▷ generating adversarial labels
      **end for**
      **adversarial training**: $\theta \leftarrow \theta - \alpha \cdot \frac{1}{n} \sum_{i=1}^n \nabla_\theta \mathcal{L}(\tilde{x}_i, \tilde{y}_i; \theta)$  ▷ updating model parameters
    **end for**
**end for**
**Output:** model parameter $\theta$.

---

## 3.2 Adversarial Interpolation Training

We utilize adversarial interpolation for robust model training. Given a set of data, $S = \{(x_i, y_i)\}$, we first generate perturbed examples $\tilde{S} = \{(\tilde{x}_i, \tilde{y}_i)\}$ using adversarial interpolation, and then use the generated examples for model training (Goodfellow et al., 2015; Madry et al., 2018). Formally, we perform robust model training by solving the following problem:

$$\theta^* = \arg\min_\theta \mathbb{E}_{(x,y)\sim S}\Big[\mathcal{L}_\theta(\tilde{x}, \tilde{y})\Big], \tag{2}$$

where the loss is the standard cross-entropy-based loss as $\mathcal{L}_\theta(x, y) \triangleq \text{CE}(g \circ f(\tilde{x}), \tilde{y})$, with $f$ the feature extraction network shared with the adversarial interpolation process and $g$ denotes some additional layers following $f$ (*e.g.*, FC layer followed by softmax layer). Here we overload $\theta$ to represent the union of parameters for $g$ and $f$. The training samples $(\tilde{x}, \tilde{y})$ are generated via adversarial interpolation as follows:

$$\tilde{x}^* = \arg\min_{\tilde{x} \in B(x,\epsilon)} \mathcal{D}(\tilde{x}, x'), \tag{3}$$

$$\tilde{y}^* = \arg\max_{\tilde{y} \in B(y,\epsilon_y)} \mathcal{D}_y(\tilde{y}, y') \tag{4}$$

which perturbs image $x$ and $y$ towards different (or reverse) directions within a permissible neighborhood. $\mathcal{D}$ and $\mathcal{D}_y$ are proper distance measures for image and label respectively. By perturbing the image and label in directions reverse to each other, the new image-label pair $(\tilde{x}, \tilde{y})$ will potentially induce a higher classification loss when evaluated together and represent a strong adversarial sample to the model (Wang & Zhang, 2019). Here $\epsilon_y$ denotes the maximum of the allowed perturbation for the label and it will be properly set to ensure the entry corresponding to the ground-truth label will remain to be the largest one after perturbation. All the samples in $B(y, \epsilon_y)$ are assumed to have non-negative elements and the entries of each sample sum to one. We reformulate Eqn.(4) using the following approximate instantiation

$$\tilde{y}^* = \arg\min_{\tilde{y} \in B(y,\epsilon_y)} \|\tilde{y} - \bar{y}'\|_2^2 \tag{5}$$

where Euclidean distance is used and $\bar{y}' = \frac{1-y'}{C-1}$ with $y'$ the label of $x'$, with $C$ the total number of classes. Note that labels are in vector form thus $y$ and $y'$ are one-hot vectors. More interpretations on Eqn.(5) are provided in Section 3.3. Based on Eqn.(3) and Eqn.(5), we can derive the updates as follows. For two points $(x, y)$ and $(x', y')$, we generate the perturbed training sample as follows

$$\tilde{x} = x - \epsilon \cdot \frac{\partial \mathcal{D}(x, x')}{\partial x} \tag{6}$$

$$\tilde{y} = y - \epsilon_y \cdot \frac{\partial \|y - \bar{y}'\|_2^2}{\partial y} = (1 - \epsilon_y)y + \epsilon_y \bar{y}'. \tag{7}$$

Eqn.(6) can be interpreted as interpolating point $x$ towards $x'$ for the data, while interpolating the label $y$ away from $y'$. The training procedure is summarized in Algorithm 1 and the implementation is

in Section A.1. In practice, $\mathcal{D}(x, x')$ is implemented using deepnet-based feature extractor combined with a standard distance measure such as Euclidean (or other metrics such as Cosine distance),

$$\mathcal{D}_\theta(x, x') = \|f_\theta(x) - f_\theta(x')\|_2^2. \tag{8}$$

We have used $\mathcal{D}$ as a shorthand for $\mathcal{D}_\theta$ in the above but without explicitly specifying the dependency on $\theta$. Minimizing this term makes the feature thus the perturbed input image incorporate some features from the other sample, while retaining visually similar to the original image, thus acting as an effective form of adversarial perturbation. More importantly, we perturb the image *towards* a target instance, while perturb the label *against* this instance (to all other classes but this target one). When the target instance is of different class with the current instance, the label interpolation operation essentially reinforces that the perturbed image is not from the target class (*c.f.* Table 7), which helps to further break the correlation between $\delta$ and $y'$, beyond decreasing the correlation between $\delta$ and $y$. When the target instance is of the same class with the current instance, the label interpolation operation is equivalent to conventional label smoothing (proof is deferred to Appendix A.3).

### 3.3 INTERPRETING AND UNDERSTANDING OF THE PROPOSED APPROACH

From the perspective of robust and non-robust features, the key to robust model training is to effectively prevent the classifier from learning useful but non-robust features (Ilyas et al., 2019). In order to achieve this, we need a way to convey this preference to the learning algorithm. Currently, one of the most straightforward way to communicate this is by manipulating the data. In fact, by manipulate the data in different ways, we can convey different preferences thus receiving different results from the machine learning algorithm (*c.f.* Table 7). As our major objective is to obtain a robust model, this can be achieved by breaking the correlation between the adversarial perturbation $\delta$ and label. Conventionally, it is achieved by constructing $\delta \sim \Delta$, where $\Delta$ is the set of non-robust features of all the other classes. In this way, by training the model using the original label $y$ and perturbation $\delta$ from all other classes, the correlation between $\delta$ and $y$ is reduced. Conversely, if we construct $\delta$ as from one class ($y'$) and use $y'$ as the label for training, then we essentially imply that $\delta$ represents useful feature for the prediction of $y'$. Therefore the model will learn to use these non-robust features for prediction, which has been exploited by Ilyas et al. (2019) recently. The proposed approach can be interpreted from two complementary perspectives. Firstly, we also de-emphasize $\delta$ *as a family* by constructing it from different classes while retaining the original target specified by $y$, thus reducing the correlation between $\delta$ and $y$. Secondly, we further de-emphasize *each individual* $\delta$ by reinforcing that $\delta$ is not useful features for $y'$, as represented by $(1-y')$, ("not $y'$"), thus further breaking the correlations between $\delta$ and $y'$. While we present one concrete instantiation in this work, the reasoning is general and can be implemented with other configurations as well.

### 3.4 CONNECTIONS WITH PREVIOUS WORKS

A feature-scattering approach for generating adversarial perturbations for training has been proposed recently (Zhang & Wang, 2019), advocating the usage of unsupervised adversarial generation to avoid label leaking. Intuitively, it generates perturbations by *moving away from a set of similar samples*, discovered through an additional matching process. Differently, adversarial-interpolation generates adversarial examples by *moving towards a distinct sample*, which does not require the expensive matching step and is complementary to feature-scattering (Zhang & Wang, 2019).

There is also an interesting connection with Mixup (Zhang et al., 2018). Specifically, in the case of $f_\theta(x) = x$ with Euclidean distance, adversarial interpolation for images (Eqn.(6)) is equivalent to the operation applied to image in Mixup (Zhang et al., 2018) (proof deferred to Section A.4). Therefore our approach can be interpreted as feature space induced adverse image generation instead of linearly mixing the raw pixels directly. Manifold-mixup (Verma et al., 2019) has been proposed recently by generalizing the mixup operation from input to latent features. While it also has an interpolation step in feature space, its motivation is different from ours and it is used differently. Manifold-mixup performs feature interpolation to generate the mixed feature, which is then passed to the subsequent layers till the cross-entropy loss. The cross-entropy loss-induced gradients backpropagate through the interpolation for model training. Manifold-mixup plays the role of a regularizer in order to obtain neural networks with smoother decision boundaries at multiple levels of representation, which helps but does not significantly improve robustness against multi-step PGD attacks (Verma et al., 2019) simiar to the original Mixup (Zhang et al., 2018). The authors of Verma et al. (2019) have further integrated Manifold-mixup with adversarial training to mitigate this issue in another recent work (Lamb et al., 2019). Different from these works, the proposed method uses features interpolation for *inducing input perturbation*, which is done by backpropagating the gradient from feature

space distance Eqn.(8) to the input (Figure 1). The perturbed inputs are then used as new inputs in place of the original ones for model training. Furthermore, empirically different from Mixup (Zhang et al., 2018) and Manifold-mixup (Verma et al., 2019), the proposed adversarial image interpolation scheme clearly helps to improve model robustness as shown in Table 5. Another crucial difference with Mixup (Zhang et al., 2018) and Manifold-mixup (Verma et al., 2019) is that the label in the proposed approach is interpolated in a direction "reverse" to that of the image interpolation to further reinforce robustness. This difference can be explained by the fact that mixup focuses more on the *on-manifold generalization* while adversarial robustness further extends the focus to *off-manifold robustness* (Stutz et al., 2019). Anonymous (2020) presents an interesting interpretation of Mixup as belonging to a class highly analogous to adversarial training. Without evaluation of adversarial robustness provided in the paper, Anonymous (2020) is mainly on interpreting and understanding of Mixup, which is different from our goal of improving model robustness. Christopher Beckham (2019) generalizes Manifold-mixup (Verma et al., 2019) for image generation, where feature interpolation is used to train a model that is able to combine the attributes of multiple inputs in a resynthesised image. The purpose of feature interpolation in (Christopher Beckham, 2019) is to introduce human sensible visual attributes in the generated image, and the image is generated by forwarding the mixed feature through a decoder in the auto-encoding framework. Differently, the purpose of feature interpolation in the proposed method is to introduce human imperceptible, non-robust features in the generated image, which is generated by back-propagating the feature difference through the feature extractor (*c.f.* Figure 1). A number of recent works that exploit interpolation from a different angle have also been presented by Wang et al. (2018b;a); Wang & Osher (2019), with the interpolation operates at the output layer and no adversarial label interpolation.

The recent Bilateral method (Wang & Zhang, 2019) also jointly perturbs image and label, but uses the classification loss directly. Our approach is very different in that the classification loss is not used in generating perturbed image or label. Instead, it uses an adversarial interpolation scheme which is not directly connected with the classification loss thus mitigating the problem of label leaking or data manifold tilting and shrinking (Tanay & Griffin, 2016; Zhang & Wang, 2019).

We have used an unsupervised approach to generate adversarial perturbations. As another form of utilizing unsupervised learning, Uesato et al. (2019) improves adversarial generalization by using additional unlabeled data, which is orthogonal to our work and could potentially be used together.

## 4 EXPERIMENTS

The `PyTorch` implementation of the proposed `Adv-Interp` approach is provided in Section A.1. We conduct extensive experiments to verify the effectiveness of the proposed approach. We compare the performance of the proposed method with a number of baseline methods:

- `Natural`: the model trained with standard approach using natural images (Krizhevsky, 2009);
- `Madry`: the PGD-based approach from Madry et al. (2018), which is one of the most effective and representative defense method;
- `Bilateral`: a method that performs adversarial training with both image and label adversarial perturbations generated using the classification loss (Wang & Zhang, 2019);
- `Feature-Scatter`: a recent method which generates adversarial perturbations using an unsupervised feature-scattering scheme for attack generation (Zhang & Wang, 2019).

Following (Madry et al., 2018), the full network (image to logits) is implemented as the Wide ResNet (WRN-28-10) (Zagoruyko & Komodakis, 2016) and $f_\theta(\cdot)$ (feature extractor) is implemented by excluding the last layer for the logits. $\mathcal{D}$ is implemented as the Cosine distance, which is free-from additional tuning parameter. For training, the initial learning rate $\alpha$ is 0.1 for CIFAR and 0.01 for SVHN. We set the number of epochs for `Natural` and `Madry` methods as 100 with transition epochs set as $\{60, 90\}$ following Madry et al. (2018); Wang & Zhang (2019). The training scheduling of 200 epochs with the same transition epochs is used similar to Wang & Zhang (2019). Standard data augmentation is used during training, *i.e.*, random crops with 4 pixels of padding and random horizontal flips (Krizhevsky, 2009). We use a perturbation budget of $\epsilon$=8 (Madry et al., 2018), $\epsilon_y$= 0.5, and $L$=1 in training. The trained models are evaluated by measuring the accuracy performance against different adversarial attacks. For *white-box* attacks, we use:

- FGSM: the Fast Gradient Sign Method (Goodfellow et al., 2015), which is a one-step gradient-based approach for generating adversarial attacks;
- PGD: Projected Gradient Descent Method (Madry et al., 2018), which is a multi-step gradient-based approach for generating adversarial attacks (PGD$T$ denotes PGD attack with $T$ iterations);

| Models | Clean | Performance under White-box Attack ($\epsilon = 8$) | | | | | Worst |
|---|---|---|---|---|---|---|---|
| | | FGSM | PGD20 | PGD100 | CW20 | CW100 | |
| Natural | 95.6 | 37.0 | 0.0 | 0.0 | 0.0 | 0.0 | 0.0 |
| Madry | 86.7 | 54.9 | 45.0 | 44.5 | 45.7 | 45.3 | 44.5 |
| Bilateral | 91.2 | 70.7 | 57.5 | 55.2 | 56.2 | 53.8 | 53.8 |
| Feature-Scatter | 90.0 | 78.4 | 70.5 | 68.6 | 62.4 | 60.6 | 60.6 |
| Adv-Interp | 90.3 | 78.0 | 73.5 | 73.0 | 69.7 | 68.7 | 68.7 |

Table 1: **CIFAR10 results**. Comparison of performance (classification accuracy) for **Natural**, **Madry** (Madry et al., 2018), **Bilateral** (Wang & Zhang, 2019), **Feature-Scatter** (Zhang & Wang, 2019) and the proposed **Adv-Interp** method under different attacks.

| Models | Performance under increasing $\epsilon$ ($8 \to 20$) with PGD | | | | | | | Performance under increasing $\epsilon$ ($8 \to 20$) with CW | | | | | | |
|---|---|---|---|---|---|---|---|---|---|---|---|---|---|---|
| | 8 | 10 | 12 | 14 | 16 | 18 | 20 | 8 | 10 | 12 | 14 | 16 | 18 | 20 |
| Natural | 0.0 | 0.0 | 0.0 | 0.0 | 0.0 | 0.0 | 0.0 | 0.0 | 0.0 | 0.0 | 0.0 | 0.0 | 0.0 | 0.0 |
| Madry | 45.0 | 34.9 | 27.0 | 20.8 | 16.8 | 13.1 | 10.1 | 45.7 | 35.6 | 27.6 | 21.4 | 16.9 | 12.9 | 9.9 |
| Adv-Interp | 73.5 | 72.9 | 71.9 | 71.3 | 70.2 | 68.6 | 67.4 | 69.7 | 68.2 | 66.6 | 65.2 | 63.8 | 62.4 | 60.5 |

Table 2: **Performance against stronger white-box attacks with increasing attack budgets.** The models are trained with the attack budget $\epsilon$=8, and are evaluated against attacks with larger budgets.

- CW: Carlini-Wagner loss based attack (Carlini & Wagner, 2017), which is also a multi-step gradient-based approach for generating adversarial attacks. It is implemented using the CW-loss (Carlini & Wagner, 2017) within the PGD framework following Madry et al. (2018).

For *black-box* attacks, we use both gradient-based and gradient-free ones as detailed in Section 4.4.

## 4.1 EVALUATION AGAINST WHITE-BOX ATTACKS ON CIFAR10

We conduct experiments on CIFAR10 (Krizhevsky, 2009) in this section. CIFAR10 is a dataset with 10 classes, 5K training images per class and 10K test images. It has been widely use in adversarial training literature (Madry et al., 2018; Wang & Zhang, 2019).

**Evaluation under Standard Setting.** We report the accuracy on the original test images (Clean) as well as under standard adversarial settings (Madry et al., 2018; Carlini & Wagner, 2017; Carlini et al., 2019) and the results are summarized in Table 1. From Table 1, we have several observations:

- it is observed that model trained with natural images (**Natural**) fails drastically under different white-box attacks. **Madry** method improves the model robustness significantly over the **Natural** model and achieves about 45.0% accuracy under the standard PGD20 attack. The **Bilateral** approach outperforms **Madry** and achieves a performance of 57.5% under PGD20 attack. The recent **Feature-Scatter** method further boost the performance to 70.5% under PGD20 attack. The proposed approach achieves 73.5% accuracy under the standard 20 steps PGD attack, outperforming all the compared methods;
- it is also noted that while the **Feature-Scatter** method achieves competitive performance under PGD20 attack, its performance has a noticeable drop when the strength of the attack is increased (*e.g.* PGD20→PGD100). The proposed **Adv-Interp** approach, on the other hand, maintains its performance when the strength of the attack is increased (PGD20→PGD100) and outperforms **Feature-Scatter** with a even large gap under PGD100 attack;
- it is interesting to observe that under the CW metric, the performances of **Madry** and **Bilateral** are similar to their respective performances under PGD attack. For **Feature-Scatter**, there is a large drop for the performance under CW attack compared with that under PGD attack. The proposed **Adv-Interp** approach, on the other hand, has a smaller performance gap between PGD and CW metric and outperforms **Feature-Scatter** and all other methods with a large margin;
- in terms of the *worst case* performance among all evaluation criteria, the proposed **Adv-Interp** method improves over **Madry** by 24.2%, **Bilateral** by 14.9% and **Feature-Scatter** by 8.1%.

| Models | Performance under increasing $T$ ($20 \to 1000$) with PGD | | | | | | | Performance under increasing $T$ ($20 \to 1000$) with CW | | | | | | |
|---|---|---|---|---|---|---|---|---|---|---|---|---|---|---|
| | 20 | 50 | 100 | 200 | 300 | 500 | 1000 | 20 | 50 | 100 | 200 | 300 | 500 | 1000 |
| Natural | 0.0 | 0.0 | 0.0 | 0.0 | 0.0 | 0.0 | 0.0 | 0.0 | 0.0 | 0.0 | 0.0 | 0.0 | 0.0 | 0.0 |
| Madry | 45.0 | 44.6 | 44.5 | 44.5 | 44.5 | 44.5 | 44.4 | 45.7 | 45.7 | 45.4 | 45.3 | 45.3 | 45.3 | 45.2 |
| Adv-Interp | 73.5 | 73.0 | 73.0 | 73.0 | 72.9 | 72.8 | 72.8 | 69.7 | 69.0 | 68.7 | 68.4 | 68.3 | 68.3 | 68.3 |

Table 3: **Performance against stronger white-box attacks with increasing attack iterations.** The models are trained with the attack budget $\epsilon$=8.

| Models | Clean | Performance under White-box Attack | | | | | Worst |
|---|---|---|---|---|---|---|---|
| | | FGSM | PGD20 | PGD100 | CW20 | CW100 | |
| Natural | 78.6 | 9.7 | 0.0 | 0.0 | 0.0 | 0.0 | 0.0 |
| Madry | 59.9 | 28.5 | 22.6 | 22.2 | 23.2 | 23.1 | 22.2 |
| Bilateral | 68.2 | 60.8 | 26.7 | 25.3 | – | 22.1 | 22.1 |
| Feature-Scatter | 73.9 | 61.0 | 47.2 | 46.2 | 34.6 | 30.6 | 30.6 |
| Adv-Interp | 73.6 | 58.3 | 41.0 | 40.2 | 32.4 | 31.2 | 31.2 |

| Models | Clean | Performance under White-box Attack | | | | | Worst |
|---|---|---|---|---|---|---|---|
| | | FGSM | PGD20 | PGD100 | CW20 | CW100 | |
| Natural | 96.6 | 36.0 | 0.3 | 0.2 | 0.3 | 0.0 | 0.0 |
| Madry | 93.9 | 68.4 | 47.9 | 46.0 | 48.7 | 47.5 | 46.0 |
| Bilateral | 94.1 | 69.8 | 53.9 | 50.3 | – | 48.9 | 48.9 |
| Feature-Scatter | 96.2 | 83.5 | 62.9 | 52.0 | 61.3 | 50.8 | 50.8 |
| Adv-Interp | 94.1 | 75.6 | 65.8 | 64.0 | 63.4 | 60.4 | 60.4 |

Table 4: **More evaluation results.** Performance comparisons on (a) CIFAR100 and (b) SVHN.

| Models | Clean | Performance under White-box Attack | | | | | Worst |
|---|---|---|---|---|---|---|---|
| | | FGSM | PGD20 | PGD100 | CW20 | CW100 | |
| image-org | 95.7 | 62.4 | 11.3 | 2.3 | 16.7 | 8.7 | 2.3 |
| image-mixup | 95.4 | 79.9 | 32.4 | 19.4 | 31.8 | 23.1 | 19.4 |
| image-cls | 90.3 | 71.2 | 60.0 | 57.7 | 56.3 | 54.1 | 54.1 |
| Adv-Interp | 90.3 | 78.0 | 73.5 | 73.0 | 69.7 | 68.7 | 68.7 |

| Models | Clean | Performance under White-box Attack | | | | | Worst |
|---|---|---|---|---|---|---|---|
| | | FGSM | PGD20 | PGD100 | CW20 | CW100 | |
| label-org | 91.6 | 57.3 | 38.1 | 36.5 | 39.3 | 38.0 | 36.5 |
| label-mixup | 93.3 | 65.5 | 19.4 | 13.4 | 18.8 | 14.1 | 13.4 |
| label-smooth | 91.1 | 77.0 | 68.4 | 66.2 | 62.9 | 61.3 | 61.3 |
| Adv-Interp | 90.3 | 78.0 | 73.5 | 73.0 | 69.7 | 68.7 | 68.7 |

Table 5: **Ablation studies** on different perturbation schemes for (a) image and (b) label.

**Evaluation against Stronger White-box Attacks.** All the robust models are trained under the standard setting with a maximum attack budget $\epsilon = 8$. While it is standard to restrict the maximum budget at test-time to be $\epsilon = 8$ as well, it is also illuminating to conduct evaluations beyond it for a comprehensive understanding of the model robustness. We therefore evaluate model robustness against PGD20 and CW20 attacks with increasing attack budgets and the results are summarized in Table 2. Unsurprisingly, the performance of **Natural** model remains at 0. **Madry** improves the model robustness significantly over **Natural** across a wide range of attack budgets beyond 8, demonstrating the effectiveness of the standard adversarial training on improving model robustness. The proposed **Adv-Interp** approach further improves the performance over **Madry** by a large margin for all the evaluated attack budgets. It is interesting to note that **Adv-Interp** maintains a high level of robustness even beyond the regime it is trained on. Note that if the attack budget is further increased to $\epsilon = 255$, the performance of all methods will be eventually approaching 0.

We further conduct another set of experiments towards evaluation against stronger attacks, by increasing the number of attack iterations with a fixed attack budget of 8. The results are summarized in Table 3. As can be observed from Table 3, **Madry** method performs stable when the number of attack iterations is increased. The proposed **Adv-Interp** method can also maintain a fairly stable performance across number of attack iterations is increased. It is interesting to note that the performance of **Adv-Interp** under PGD1000 still outperforms the performance of all other approaches under PGD20 by a large margin (*c.f.* Table 1). We will use a PGD/CW attackers with $\epsilon = 8$ and attack step 20 and 100 in the sequel as the standard setting for evaluating model robustness.

## 4.2 WHITE-BOX ATTACK EVALUATION ON MORE DATASETS

**CIFAR100.** We also experiment with CIFAR100 dataset, with 100 classes, 50K training and 10K test images (Krizhevsky, 2009). This data set is more challenging as the number of training images per class is much smaller (Zhang & Wang, 2019). We set training epoch as 300 and $\epsilon_y = 0.9$. As shown by the results in Table 4(a), the proposed approach outperforms **Madry** and **Bilateral** by a large margin under both PGD and CW attacks. Furthermore, it achieves performance on-par with **Feature-Scatter** under the strongest attack (CW100) on this challenging dataset.

**SVHN.** We further report results on the SVHN dataset (Netzer et al., 2011), which is a 10-way house number classification dataset, with 73257 training images and 26032 test images. The additional training images are not used in experiment. The results are summarized in Table 4(b). Experimental results show that the proposed method outperforms other baseline methods with a clear margin under both PGD and CW attacks. Notably, it outperforms **Feature-Scatter** by about 10% under the worst case metric, further demonstrating the effectiveness of the proposed approach.

## 4.3 ABLATION STUDIES: THE IMPACTS OF ADVERSARIAL INTERPOLATION

We conduct a number of ablation studies to investigate the impacts of the components in the adversarial interpolation scheme. We replace a single component while keeping all others the same. **Impacts of Adversarial Image Interpolation.** We investigate the impacts of image adversarial interpolation by comparing with the following: *i*) **image-org**: use the original image for training; *ii*) **image-mixup**: use the mixup operation for generating perturbed images (Eqn.(19)); *iii*) **image-cls**: use classification loss for generating perturbed image; The results are presented in Table 5(a). It is observed that there is a significant performance drop when removing image adversarial interpolation (**image-org**). **image-mixup** performs better than **image-org** with improved robustness. **image-cls** is more effective due to the usage of classification loss in generating adver-

| Attack Gradient | Performance under Black-box Attack | | | | | Gradient-Free | Performance under |
|---|---|---|---|---|---|---|---|
| Generation Models | PGD20 | PGD100 | CW20 | CW100 | | Attack Methods | Black-box Attack |
| **Natural** | 89.3 | 89.1 | 89.4 | 89.2 | | **SPSA** | 89.2 |
| **Adv-Interp'** | 82.9 | 82.5 | 81.8 | 82.0 | | **Local-Search** | 90.1 |

Table 6: **Black-box attack evaluations.** Performance of the model trained with the **Adv-Interp** method under (a) *gradient-based* and (b) *gradient-free* black-box attacks.

sarial images. The proposed method, which uses adversarial interpolation for generating perturbed images, achieves the best performance, implying the importance of adversarial image interpolation. **Impacts of Adversarial Label Interpolation.** Similarly, we replace the adversarial label interpolation part with different approaches: *i*) **label-org**: the original label is used for training; *ii*) **label-mixup**: a mixup operation is applied to the label (Eqn.(20)); *iii*) **label-smooth**: label smoothing is applied to the label (Eqn.(16)); The results are summarized in Table 5(b). It is observed that, without adversarial interpolation for label (**label-org**), the performance drops significantly. **label-mixup** improves performance under weak attacks (FGSM), but the performance under stronger attacks is decreased, possibility due to its inability in de-emphasizing the correlations between the perturbation and $y'$. **label-smooth** achieves a much better performance than **label-org** and **label-mixup**. The **Adv-Interp** method further boosts the performance over **label-smooth**, demonstrating the importance and effectiveness of adversarial label interpolation.

### 4.4 PERFORMANCE UNDER BLACK-BOX ATTACKS

To further verify that the improvement is due to inherent improvement in model robustness instead of a degenerate solution, we evaluate the robustness of the model trained with the proposed approach *w.r.t.black-box* attacks following previous evaluation setup (Tramèr et al., 2018; Uesato et al., 2018).

**Gradient-based Black-box Attack.** We first use gradient-based approach for generating the attacks, but using models that are different from the one to be evaluated. Specifically, we use the **Natural** model and **Adv-Interp'**, which is another model trained with the proposed method in a different training session. The results are summarized in Table 6 (a). It is observed that the model trained with the proposed approach is robust against different types of gradient-based black-box attacks.

**Gradient-free Black-box Attack.** We further conducted evaluation using two gradient-free black-box attacks. **SPSA** is a technique developed for high-dimensional optimization problem (Spall, 1992) and is used in Uesato et al. (2018) for gradient-free attack-based evaluations. **Local-Search** is a local search-based black-box attack method (Narodytska & Kasiviswanathan, 2017). The evaluation results of the proposed model against gradient-free attacks are summarized in Table 6 (b). All these results together with the white-box attack results in Table 1 verify that the improvements are indeed due to the improved model robustness, not because of gradient-masking caused by a degenerate solution (Tramèr et al., 2018).

## 5 CONCLUSION AND FUTURE WORK

An adversarial interpolation training method is presented in this paper. This approach is intuitive to understand and simple to implement, yet achieving state-of-the-art performance. Extensive experiments on several standard datasets have been done to evaluate and understand the performance of the proposed method. Empirical results demonstrate that the proposed approach compares favorably with existing methods in the literature. There are some limitations of the proposed approach due to the usage of an unsupervised surrogate function for generating image perturbations. While this offers the potential of mitigating some issues of the discriminative perturnation, the perturbations generated this way are not necessarily the most effective ones due to potential mis-alignment with the downstream task. Also, as there is no explicit force for further encouraging the clustering of features and the increasing of the separation margins between them, the feature points are typically more scattered (*c.f.* Figure 3), offering more opportunity to feature space attacks. Nevertheless, the fact that such a simple approach can achieve encouraging performance suggests that this is a valuable direction to explore further and the observation that it behaves differently with existing methods against different attacks suggests the potential to combine their complementary strength for further improvements (Pang et al., 2019). The proposed method is generally applicable can potentially be extended to applications beyond image classification. Furthermore, as the proposed adversarial interpolation-based attack generation does not require label information, it would be interesting to extend this approach to other types of learning tasks such as unsupervised learning and reinforcement learning. We leave the exploration of these interesting directions as future work.

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

## A APPENDIX

### A.1 ADVERSARIAL INTERPOLATION TRAINING CODE

Here we provide the code snippet for one epoch of adversarial interpolation training in **PyTorch**:

```
for x, y in data_loader:
    x_tilde, y_tilde = adv_interp(x, y, net, num_classes)
    logits_tilde = net(x_tilde, mode="logits")
    loss = soft_ce_loss(logits_tilde, y_tilde)
    optimizer.zero_grad()
    loss.backward()
    optimizer.step()

def adv_interp(x, y, net, num_classes,
               epsilon=8, epsilon_y=0.5, v_min=0, v_max=255):
    # x: image batch with shape [batch_size, c, h, w]
    # y: one-hot label batch with shape [batch_size, num_classes]
```

```
13     inv_index = torch.arange(x.size(0)-1, -1, -1).long()
14     x_prime = x[inv_index, :, :, :].detach()
15     y_prime = y[inv_index, :]
16     x_init = x.detach()+torch.zeros_like(x).uniform_(-epsilon, epsilon)
17     x_init.requires_grad_()
18     _net = copy.deepcopy(net)
19     _net.eval()
20     loss_adv = cos_dist(_net(x_init, mode="feature"),
21                         _net(x_prime, mode="feature"))
22     _net.zero_grad()
23     loss_adv.backward()
24     x_tilde = x_init.data - epsilon * torch.sign(x_init.grad.data)
25     x_tilde = torch.min(torch.max(x_tilde, x - epsilon), x + epsilon)
26     x_tilde = torch.clamp(x_tilde, v_min, v_max)
27     y_bar_prime = (1 - y_prime) / (num_classes - 1)
28     y_tilde = (1 - epsilon_y) * y + epsilon_y * y_bar_prime
29     return x_tilde.detach(), y_tilde.detach()
```

## A.2 DERIVATION OF THE EQUIVALENT LOSS FOR ADVERSARIAL LABEL INTERPOLATION

Our label interpolation procedure is implemented as

$$\tilde{y}^* = \arg \min_{\tilde{y} \in B(y,\epsilon_y)} \|\tilde{y} - \bar{y}'\|_2^2, \tag{9}$$

which leads to the following update equation for label

$$\tilde{y} = y - \epsilon_y \frac{\partial \|y - \bar{y}'\|_2^2}{\partial y} = (1 - \epsilon_y)y + \epsilon_y \bar{y}' \tag{10}$$

where the label target $\bar{y}'$ is constructed as $\bar{y}' = \frac{1-y'}{C-1}$ with $C$ the total number of classes. Here we show that this can be derived from a complementary perspective as follows.

Given a training sample $(x, y)$, the original way for model training is as follows:

$$\min_\theta \mathrm{CE}(x, y) = \min_\theta \sum_i y_i h_\theta(x)_i, \tag{11}$$

where $z = h(x) = g \circ f(x)$.

By adding a perturbation $\delta$ which fools the predictor towards label $y'$, we have $\tilde{x} = x + \delta$, which semantically still belongs to the original class. Therefore, we can use the following formulation to dis-emphasize $\delta$

$$\min_\theta \mathrm{CE}(\tilde{x}, y) = \min_\theta \sum_i y_i h_\theta(\tilde{x})_i. \tag{12}$$

Secondly, while the perturbed sample $\tilde{x}$ fools the predictor towards label $y'$, $\tilde{x}$ is *not* a valid sample of class $y'$, which is represented as

$$\min_\theta \mathrm{CE}(\tilde{x}, \bar{y}') = \min_\theta \sum_i \bar{y}_i h_\theta(\tilde{x})_i. \tag{13}$$

By combining Eqn.(12) with Eqn.(14), we have

$$\min_\theta (1 - \epsilon_y)\mathrm{CE}(\tilde{x}, y) + \epsilon_y \mathrm{CE}(\tilde{x}, \bar{y}') = \min_\theta \mathrm{CE}(\tilde{x}, (1 - \epsilon_y)y + \epsilon_y \bar{y}'), \tag{14}$$

which leads to the same update equation for label as in Eqn.(10).

## A.3 CONNECTION OF ADVERSARIAL LABEL INTERPOLATION WITH LABEL SMOOTHING

Here we provide the proof that when the target instance is of the same class with the current instance, *i.e.*, $y = y'$, the label interpolation operation is equivalent to conventional label smoothing.

Recall that for adversarial interpolation of label, we have

$$\tilde{y} = (1 - \epsilon_y)y + \epsilon_y \bar{y}', \tag{15}$$

where $\bar{y}' = \frac{1-y'}{C-1}$. $C$ is the number of classes. In the case that $y' = y$, we have

$$\begin{aligned}
\tilde{y} &= (1 - \epsilon_y)y + \epsilon_y \bar{y}' \\
&= (1 - \epsilon_y)y + \frac{\epsilon_y}{C-1}(1 - y') \\
&= (1 - \frac{C}{C-1}\epsilon_y)y + \frac{\epsilon_y}{C-1},
\end{aligned} \tag{16}$$

which is equivalent to label-smoothing with the strength of $\frac{C}{C-1}\epsilon_y$ (Szegedy et al., 2016).

### A.4 CONNECTION OF ADVERSARIAL IMAGE INTERPOLATION WITH MIXUP

There is a connection between the proposed method and mixup (Zhang et al., 2018) as mentioned in the main paper. Specifically, in the case of $f_\theta(x) = x$ with Euclidean distance, the adversarial interpolation for images (Eqn.(6)) is equivalent to mixup (Zhang et al., 2018). Concretely, for

$$\mathcal{D}(x, x') = \|x - x'\|_2^2 \tag{17}$$

we have its gradient as

$$\frac{\partial \mathcal{D}}{\partial x} = x - x'. \tag{18}$$

Plugging it into adversarial interpolation update Eqn.(6), we have

$$\tilde{x} = x - \epsilon(x - x') = (1 - \epsilon)x + \epsilon x'. \tag{19}$$

This shows that for image manipulation, when the proposed approach is performed at pixel level we recover mixup (Zhang et al., 2018). In this case, the new image $\tilde{x}$ is obtained as a linear combination of the data points $x$ and $x'$ in the original data space, which are pixels for images. This formally shows that the image manipulation in mixup can be interpreted as one gradient step for minimizing the pixel value distance between two samples. As for labels, mixup (Zhang et al., 2018) shares a similar update form as Eqn.(7) but with $\bar{y}' = y'$, *i.e.*

$$\tilde{y} = (1 - \epsilon_y)y + \epsilon_y y'. \tag{20}$$

### A.5 CATEGORIZATION OF DIFFERENT TRAINING METHODS

Here we provide a categorization of different training methods as in Table 7. By manipulating image and label in different ways, we can convey different messages to the learning algorithm and obtain the corresponding results. The proposed approach can be intuitively understood as leveraging two complementary aspects for improving model robustness, *i.e.*, by reducing the the correlations between $\delta$ and $y$ as well as the correlations between $\delta$ and $y'$.

| Methods | Training sample | | Message | Result |
|---|---|---|---|---|
| | image $\tilde{x}$ | label $\tilde{y}$ | | |
| standard training (Krizhevsky et al., 2012) | $x$ | $y$ | no preference over different features | learning non-robust model with high probability |
| adversarial training (Goodfellow et al., 2015) | $x + \delta$ | $y$ | prefer robust features by reducing the correlations between $\delta$-$y$ | learning robust model |
| non-robust fea. learning (Ilyas et al., 2019) | $x + \delta$ | $y'$ | prefer non-robust features by leveraging the correlations between $\delta$-$y'$ | learning non-robust model |
| this work | $x + \delta$ | $(1-\epsilon_y)y + \epsilon'_y(1-y')$ | prefer robust features by reducing the correlations between $\delta$-$y$ and $\delta$-$y'$ | learning robust model |

Table 7: **Categorization and analysis of different model training methods**. $x$ denotes the original image and $y$ denotes the original label. $\delta$ is the perturbation added to the image $x$, constructed by altering the prediction towards $y'$. $\epsilon'_y = \epsilon_y/(C-1)$ where $C$ is the total number of classes. Note that different approaches can use different methods for generating $\delta$.

## A.6 Label Adv-Interpolation Parameter

We investigate the impacts of the adversarial label interpolation parameter $\epsilon_y$ and the results are summarized in the table on the right. It is observed from the results that the proposed approach performs well for a range of values of $\epsilon_y$.

| $\epsilon_y$ | Clean | Performance under White-box Attack ($\epsilon = 8$) | | | | |
|---|---|---|---|---|---|---|
| | | FGSM | PGD20 | PGD100 | CW20 | CW100 |
| 0 | 91.6 | 57.3 | 38.1 | 36.5 | 39.3 | 38.0 |
| 0.1 | 90.9 | 73.3 | 57.3 | 52.9 | 57.2 | 54.2 |
| 0.2 | 90.5 | 74.9 | 61.6 | 58.5 | 63.4 | 61.5 |
| 0.3 | 90.6 | 78.2 | 73.5 | 72.6 | 68.7 | 68.0 |
| 0.4 | 90.8 | 77.8 | 73.7 | 73.0 | 67.7 | 66.6 |
| 0.5 | 90.3 | 78.0 | 73.5 | 73.0 | 69.7 | 68.7 |
| 0.6 | 90.5 | 74.9 | 61.6 | 58.4 | 63.5 | 61.5 |
| 0.7 | 90.2 | 73.5 | 59.8 | 48.7 | 59.2 | 50.8 |

## A.7 Investigation of the Adversarial Losses

Following Madry et al. (2018), we investigate and visualize the variations of the losses caused by the adversary (adversarial losses). We run PGD adversary with random starts for 10 times, against **Natural** model as well as the proposed **Adv-Interp** model. The results are presented in Figure 2. It can be observed that the loss achieved with PGD adversary against our model increases in a fairly consistent way and plateaus rapidly for 10 different runs with random starts, with relatively smaller variance and final loss value, compared with those of **Natural** model. These observations are consistent with properties of robust models observed in Madry et al. (2018), further verifying the inherent robustness improvement of the proposed model.

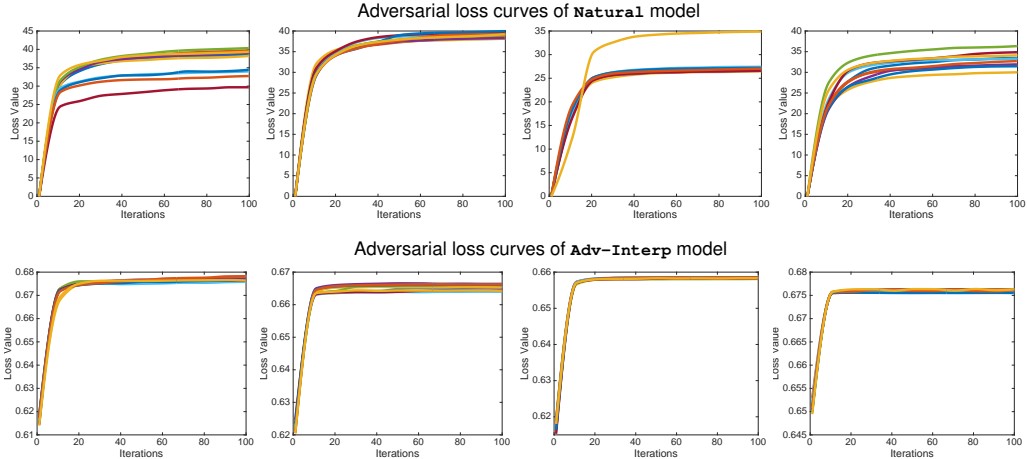

Figure 2: **Adversarial loss curves** of different models on different images under PGD attack with increasing iterations (1→100). Each column corresponds to the loss curves generated using the same input image on the **Natural** model (top) and the **Adv-Interp** model (bottom).

## A.8 Feature Distribution Visualization

We visualize the distribution of features extracted using **Natural** and **Adv-Interp** models with the t-SNE visualization technique and the results are shown in Figure 3. Specifically, we show the feature distributions for Clean image as well as perturbed images generated using PGD and adversarial interpolation. It is observed that the **Natural** model changes drastically in the presence of attacks. The proposed **Adv-Interp** model, on the other hand, can better preserve the feature distribution against adversarial attacks.

## A.9 Adversarial Interpolation Image Visualization

We provide the visualization of original natural images and the perturbed images generated by adversarial interpolation using the proposed model in Figure A.9. Here we conduct the visualization with different number of attack steps $L = 1, 3, \cdots$. Note that in model training, we have used $L = 1$.

## A.10 More Results and Comparisons

We have discussed the connection of the proposed approach with a number of related recent works in Section 3.4. Here we provide more comparison with them in Table 8. Specifically, we compared

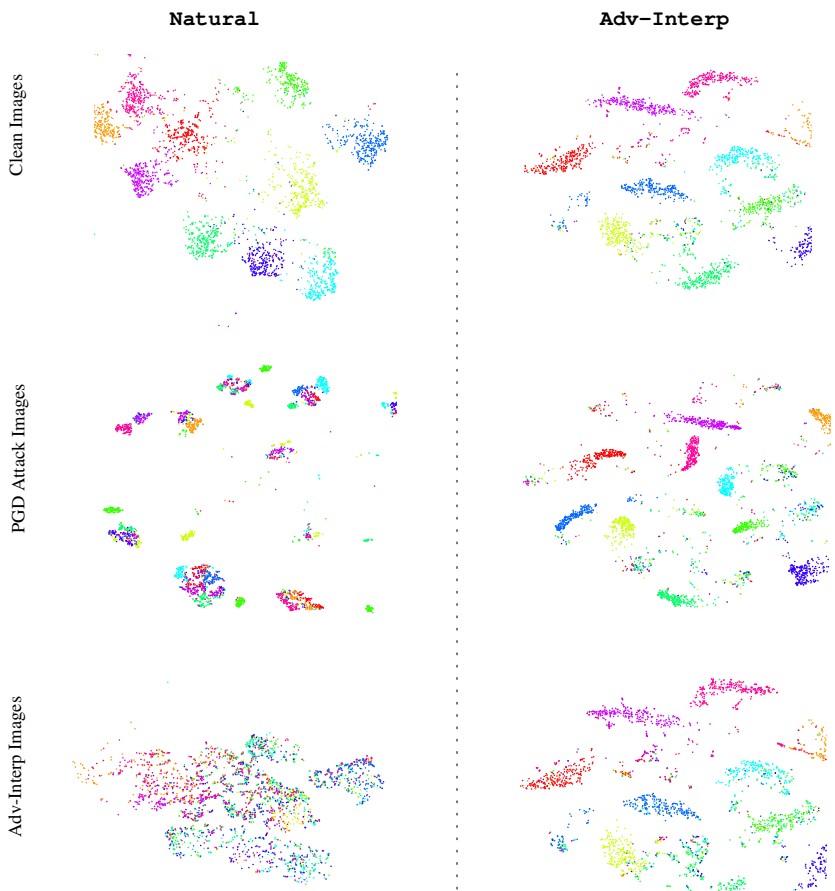

Figure 3: **Feature distribution visualization** of `Natural` and `Adv-Interp` models on (top) clean images, (middle) PGD attack perturbed images, and (bottom) adversarial interpolation perturbed images. It is observed that the distribution of features extracted using the `Natural` model changes drastically in the presence of attacks. On the other hand, the `Adv-Interp` model can better preserve the feature distribution against attacks.

with `Mixup` (Zhang et al., 2018), `Manifold-mixup` (Verma et al., 2019) as well as their adversarial extensions `IAT-Mixup` and `IAT-Manifold-mixup` (Lamb et al., 2019), and `UAT` (Uesato et al., 2019), which is an adversarial training method augmented with an unsupervised loss computed on additional unlabeled data. The results in Table 8 for `UAT` is obtained using 20k additional images per class for training (Uesato et al., 2019).

| Attack Methods | Models | | | | | |
|---|---|---|---|---|---|---|
| | Mixup | Manifold-mixup | IAT-Mixup | IAT-Manifold-mixup | UAT | Adv-Interp |
| Natural | 96.8 | 96.9 | 93.6 | 93.5 | 85.9 | 90.3 |
| FGSM | 67.4 | 61.6 | 66.2 | 64.8 | - | 78.0 |
| PGD20 | 0.7 | 1.7 | 50.1 | 44.8 | 62.2 | 73.5 |

Table 8: **More results** and comparisons with several recent methods on CIFAR10.

## A.11 EVALUATIONS AGAINST ADAPTIVE ATTACKS

Here we provide more evaluations against adaptive attacks following reviewer's suggestions as well as Carlini et al. (2019). We include the following aspects of "adaptiveness" in our evaluations: 1) we assume the full knowledge of the model as well as other necessary information and perfrom evaluations with multiple runs; 2) we include feature-based attack, which has the full knowledge of the model as well as the defense method. Specifically, we use the following attacks for evaluation:

- **PGD-min**: we run the evaluation agasint PGD20 attack 10 times with different random starts for the same test sample and record the worst performance for each sample; the average performance over the whole test set is reported as the final result.

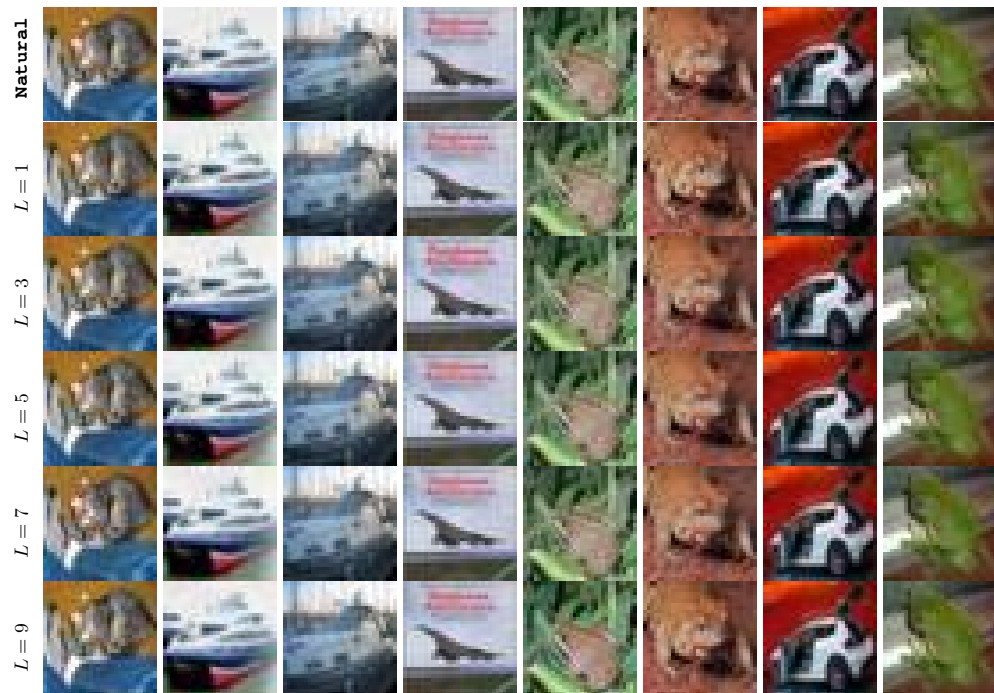

Figure 4: **Visualization of original natural images and the perturbed images** generated by adversarial interpolation with different number of attack steps ($L$) for the perturbation budget of $\epsilon = 8$.

- **CW-min**: we run the evaluation agasint CW20 attack 10 times with different random starts for the same test sample and record the worst performance for each sample;
- **Feature-min** attack: we run feature attack 10 times by using different random starts, different target images and 20 attack steps, and record the worst performance for each sample.

We have also provided the results on original clean images for reference. We conducted evaluations on `Natural`, `Madry`, and `Adv-Interp`. `Adv-Interp*` denotes the model with exactly the same trained parameters as `Adv-Interp`, but only add a random perturbation to the input.[1] The results are shown in Table 9. We have the following observations from the results. The `Adv-Interp` perorms well agasint PGD-min and CW-min attacks with multiple random starts, suggesting its robustness against these attacks. It performs less favorably against `Madry` to Feature attack, but the gap is further bridged by `Adv-Interp*`. More specifically, `Adv-Interp*` still achieves higher performance against PGD-min and CW-min attacks, and performs comparably to `Madry` on Feature-min attack. Essentially, the random perturbation does not impact the effectiveness of PGD-min and CW-min attacks (as expected), while it shows more impacts on Feature-min attack. These results imply that Feature attack is a less generic type of attack and generally less robust as it could be easily affected by (typically non-impactful) random perturbations. The results also suggest that proposed model could potentially benefit further by improving the surrogate function, *e.g.*, by incorporating explicit terms for enlarging the separation margin between classes, which is an interesting direction that is worthwhile to be explored further in the future.

| Attack Methods | Models | | | |
|---|---|---|---|---|
| | `Natural` | `Madry` | `Adv-Interp` | `Adv-Interp*` |
| Clean | 95.6 | 86.7 | 90.3 | 90.1 |
| PGD-min | 0.0 | 44.5 | 71.5 | 70.0 |
| CW-min | 0.0 | 45.2 | 67.2 | 65.4 |
| Feature-min | 0.0 | 42.3 | 36.4 | 43.4 |

Table 9: **More results** against adaptive attacks on CIFAR10.

---

[1] $x \rightarrow \mathrm{clamp}(x + \epsilon \cdot \mathrm{rand\_sign\_like}(x), 0, 255)$, where $\mathrm{rand\_sign\_like}(x)$ returns a tensor with the same shape as $x$ and the elements are randomly set to $-1$ or $1$. $\mathrm{clamp}$ ensures the input is within the specified range. $\epsilon$ denotes the perturbation budget which is the same as used in training.

