# OpenReview forum: "Adversarial Interpolation Training: A Simple Approach for Improving Model Robustness"
_ICLR.cc/2020/Conference — Reject_

### Official Review · AnonReviewer1 · 2019-10-14
**Official Blind Review #1**

**Rating:** 6

**Review:**

Contribution: This paper proposed an adversarial interpolation approach for generating adversarial samples and then training robust deep nets by leveraging the generated adversarial samples. However, I have the following concerns:

1. In adversarial interpolation training, why \tilde{y}_i' is set to 1/(C-1)(1-y_{n-i+1})?

2. This work lack of interpretation of why the proposed method is more effective than PGD adversarial training.

3. How about training deep nets with replicas of the training data but replace the true labels with random labels? I want to see such a result.

4. Can the authors provide the black-box attack results also? I want to see the performance of PGD adversarially trained deep nets on the adversarial images crafted by attacking Adv-Interp trained deep nets, and vice versa.

5. Can the authors provide the visualization of a few adversarial interpolated images?

6. The authors should compare with the existing efforts that using interpolation to improve adversarial robustness. Below are some Related works on using interpolation in deep nets to improve their robustness

1). Bao Wang, Xiyang Luo, Zhen Li, Wei Zhu, Zuoqiang Shi, Stanley J. Osher. Deep Neural Nets with Interpolating Function as Output Activation, NeurIPS, 2018

2). Bao Wang, Alex T. Lin, Zuoqiang Shi, Wei Zhu, Penghang Yin, Andrea L. Bertozzi, Stanley J. Osher. Adversarial Defense via Data Dependent Activation Function and Total Variation Minimization, arXiv:1809.08516, 2018

3). B. Wang, S. Osher. Graph Interpolating Activation Improves Both Natural and Robust Accuracies in Data-Efficient Deep Learning, arXiv:1907.06800, 2019

7. Moreover, the following paper provides a theoretical interpretation of adversarial vulnerability of deep nets, and proposed a nice ensemble of neural SDEs to improve deep nets' robustness.

1). Bao Wang, Binjie Yuan, Zuoqiang Shi, Stanley J. Osher. ResNets Ensemble via the Feynman-Kac Formalism to Improve Natural and Robust Accuracies, arXiv:1811.10745, NeurIPS, 2019


Please address the previously mentioned concerns in rebuttal, and this paper can be acceptable if all my concerns are addressed.

**Experience Assessment:**

I have published in this field for several years.

**Review Assessment: Checking Correctness Of Derivations And Theory:**

I carefully checked the derivations and theory.

**Review Assessment: Checking Correctness Of Experiments:**

I carefully checked the experiments.

**Review Assessment: Thoroughness In Paper Reading:**

I read the paper thoroughly.

---

> ### Author Response · Authors · 2019-11-08
> **Thanks for your comments and we have addressed you comments below and in revised paper**
>
> A1: It is emerged in Eqn.(5) as an approximate reformulation to Eqn.(4). In Eqn.(4), we are trying to move away from the target label $y'$ by \emph{maximizing} the distance wrt it. In Eqn.(5), we approach the goal of move away from the target label $y_i'$ by \emph{minimizing} the distance wrt all labels except the target label $y_i'$, i.e. $\bar{y}_i'=\frac{1}{C-1} (1-y_i')$, where $y'_i$ is the one-hot representation of label. $\frac{1}{C-1}$ is a normalization term to ensure all elements in $\bar{y}'_i$ are sum to one.  Finally, in Algorithm 1, for practical implementation, we set $y_i'=y_{n-i+1}$, which essentially sets another sample from the same batch as the target sample. This is just one concrete practical implementation which introduces minimum computational overhead. Other choices such as selecting target samples from another batch are also possible.
>
> A2: We can explain the effectiveness of the proposed method from two aspects:
> 1) the fact that the proposed method exploits two complementary aspects for improving model robustness is its first advantage over conventional PGD adversarial training, as presented in Section 3.3, Section A.5 and Table 7.
> Conventional PGD adversarial training reduces correlation between additive perturbation $\delta$ and $y$  by constructing $\delta\!\!\sim\!\!\Delta$, where $\Delta$ is the set of non-robust features of all the other classes.
> The proposed approach not only reduces the the correlations
> between $\delta$ and $y$ (as in conventional PGD adversarial training) but also reduces the correlations between $\delta$ and $y'$.
> 2) the usage of unsupervised adversarial perturbation is another advantage of the proposed method.
> Conventional adversarial examples are \emph{decision boundary oriented} due to the direct usage of the cross-entropy loss, making the effective  manifold for training deviate from the original due to  \emph{tilting} and \emph{shrinking},  suffering from problems such as label leaking.
> The proposed approach uses an adversarial interpolation scheme which
> is not directly connected with the classification loss thus mitigating the problem of label leaking or data manifold tilting and shrinking, following a similar line of reasoning with feature-scattering (as mentioned in Section 3.4).
> We plan to conduct rigorous theoretical analysis of the proposed model as  next step.
>
> A3: We conduct model training as suggested. Specifically, for an original training batch $\{(x_i, y_i)\}$, we construct a new  training batch as $\{(x_i, y_i), (x_i, y_i')\}$, where $y_i'$ is a random label.
> The results are shown below.
>
> 		          Model             | Natural | FGSM  | PGD20  | PGD100  | CW20  | CW100
> 		————————————————————————————————————
> 		suggested baseline | 90.2     |   16.4    |     0.0     |       0.0     |    0.0   |    0.0
>
>
>
> A4: We have provided  evaluations against both gradient-based and gradient-free black-box attacks in Section 4.4, including the SPSA method which is also suggested by Carlini, Nicholas, et al in the paper "On evaluating adversarial robustness." arXiv preprint arXiv:1902.06705 (2019).
> We also conduct additional evaluations according to the reviewer's suggestion and the results are summarized below. All these black-box evaluations, together with the worst case white-box performance  (68.7), jointly verify the improved performance is due to the inherent improvement of model robustness instead of gradient-masking.
>
> 		Model       |  Natural | PGD  | Proposed
>                ———————————————————
> 		Natural     |     0.0     |	80.9 | 	88.5
> 		  PGD         |    85.8    | 	44.5 |	83.0
> 		Proposed |    89.1     | 71.3 |     73.0
>
>
> A5: Thanks for the suggestion! We have added the visualization of adversarial interpolated images in the revised paper (Section A.9).
>
>
> A6: Thanks for the related references. We have added discussion and citations to them in the revised paper.
> The methods presented in these works (e.g. weighted nonlocal Laplacian (WNLL) method by Wang et al 2018) are interesting. Although both with the term of interpolation, there are some crucial differences with ours.
>
> One big difference is that the proposed method uses latent feature interpolation to induce perturbation at input layer, while the WNLL method (Wang et. al 2018) exclusively operates at the output layer.
> Another difference is that in our method, the images and labels are perturbed simultaneously. Specifically, the image is interpolated towards a target image, while the label is away from the target label. WNLL does not have similar adversarial label interpolation procedure.
>
> We view WNLL and the proposed method as different approaches that exploit the 'interpolation' idea from different angles.
>
>
> A7: Thanks for the reference, which provides a nice theoretical interpretation of adversarial vulnerability of deep nets. We appreciate the contribution of this work and will cite it in our revised paper.

---

> > ### Comment · AnonReviewer1 · 2019-11-14
> > **Thank you for your reply**
> >
> > I have read your reply, and most of my questions are addressed. Accordingly, I raise my rating to weak accept. However, please also add some discussion in your revised paper about when interpolation will be helpful. As pointed out by the public comment, the proposed method is not very robust to feature attack.

---

> > > ### Author Response · Authors · 2019-11-15
> > > **Thanks again and great to know that your concerns have been addressed**
> > >
> > > Thanks for your reply and it is great to know that your concerns have been addressed!
> > >
> > > Following your further suggestion, we have added discussion in Section 5 of the revised paper on the limitations of the proposed approach and potential further improvements.
> > > The fact that such a simple approach can achieve encouraging performance suggests that this is a potentially valuable direction to explore further. And we have released the full implementation of our approach as well as the trained model ( https://github.com/Adv-Interp/adv_interp ) to contribute to the community and benefit fellow researchers.
> > >
> > > Thanks again for recognizing our contribution and for your valuable help in improving our work!

---

### Official Review · AnonReviewer3 · 2019-10-21
**Official Blind Review #3**

**Rating:** 3

**Review:**

This is an interesting work proposing a new robust training method using the adversarial example generated from adversarial interpolation. The experimental results seem surprisingly promising. The ablation studies show that both image and label interpolating help the robustness improvement.

I think it is important to provide a running time comparison between the proposed method and SOTA robust training method such as Madry's. Since the feature extractor is implemented by excluding the last layer for the logits, the backprop goes through almost the entire model. It seems that the proposed interpolating method has a similar amount of computation as PGD, so the training should take similar time as Madry's if it can converge quickly.

Also, there are too many works on robustness defense that have been proven ineffective (consider the works by Carlini). Since this is a new way of robust training and there is no certified guarantee, I would be very conservative and suggest the authors refer the checklist in [1] to evaluate the effectiveness of the defense more thoroughly to convince the readers that it really works. Especially, a robustness evaluation under adaptive attacks is necessary. In other words, if the attacker knows the strategy used by the defender, it may be possible to break the model. PGD and CW are non-adaptive since no defender information is provided.

[1] Carlini, Nicholas, et al. "On evaluating adversarial robustness." arXiv preprint arXiv:1902.06705 (2019).

**Experience Assessment:**

I have published one or two papers in this area.

**Review Assessment: Checking Correctness Of Derivations And Theory:**

I assessed the sensibility of the derivations and theory.

**Review Assessment: Checking Correctness Of Experiments:**

I assessed the sensibility of the experiments.

**Review Assessment: Thoroughness In Paper Reading:**

I read the paper thoroughly.

---

> ### Author Response · Authors · 2019-11-08
> **Thank you for sharing your comments! We have addressed them below and in the updated paper**
>
> A1: Thanks for the suggestion.  The number of attack iterations is set as $L\!\!=\!\!1$ in our training. We apologize for not making this point clear and have improved it in our revised paper (Section 4).
> One-step adversary is used for training in our model as shown in the code in Section A.1 (and now in Section 4 as well), i.e., the number of iterations $L$ in Algorithm 1 is set as $L\!\!=\!\!1$.
> As a comparison, $L\!\!=\!\!7$ is typically used in the conventional PGD adversarial training (e.g. Madry).
> Therefore, although the computational cost is comparable for a single adversary step as your correctly mentioned, the proposed method has a smaller computational cost due to the usage of single step adversary compared to conventional multi-step PGD training. We have added explanation of $L$ to the experimental details in Section 4 according to your suggestion.
>
>
> A2: Thanks for your great suggestion! We fully agree with you that we should view any new defense methods from a conservative perspective and need to be careful about evaluating them. We are indeed sharing the same perspective as you and have made the following efforts towards it:
>
> (1) We have evaluated our model from different aspects and against different kinds of attacks, including different white-box attacks, gradient-based/gradient-free black-box attacks in Section 4, which essentially follows the suggestion in the suggested paper by Carlini, Nicholas, et al. "On evaluating adversarial robustness".
> (2) We have also conducted diagnosis analysis on the adversarial loss following Madry et al. 2018 [1] (Section A.7).
> (3) Furthermore, following your suggestion, we have added one new section dedicated to the evaluation with adaptive attacks (Section A.11). We have gained some new insights following your valuable suggestions as detailed in the revised paper, which point to interesting directions that are worthwhile to be explored further in the future.
> (4) We are aware of the fact that the progress of adversarial defense could further benefit from interactions with the community. We have therefore provided the core algorithm in our paper and released the full implementation as well as the trained model ( https://github.com/Adv-Interp/adv_interp ) following the spirit from the suggested paper of Carlini et al., to further contribute to the community and also benefit fellow researchers.
>
>
> Reference
> [1] Aleksander Madry, Aleksandar Makelov, Ludwig Schmidt, Dimitris Tsipras, and Adrian Vladu. Towards deep learning models resistant to adversarial attacks. ICLR 2018

---

### Official Review · AnonReviewer2 · 2019-11-04
**Official Blind Review #2**

**Rating:** 3

**Review:**

The paper proposes adversarial interpolation training, which perturbs the images and labels simultaneously. The perturbed image $\tilde x$ is around $x$ and interpolated towards another image $x'$ while the corresponding $\tilde y = (1-\epsilon_y)y +\epsilon_y\frac{1-y'}{C-1}$ is near $y$ but away from $y'$. The distance of interpolating images is L2 distance in the feature space and that for labels is L2 distance in the label space. The paper provides an interpretation of the proposed approach from the perspective of robust and non-robust features. Thorough experiments on different types of attacks and different datasets are performed. Although the results are impressive, I still have some concerns on the method itself:

1. The method seems like a combination of manifold mixup [1] and adversarial training. The interpolation in the feature space is not a new idea and has been explored in Manifold Mixup [1]. The method resembles manifold mixup if we focus on $x$ because $\tilde x$ and $\tilde y$ both retain the original image and target $(x, y)$. The "adversarial" interpolation part is from $(x', y')$ in the sense that $\tilde y$ is away from $y'$.

2. The paper lacks a theoretical explanation, which makes it less convincing how it works so well.

3. I noticed several papers with similar ideas, e.g. [2,3,4]. Could you please discuss the connections with them? I also suggest adding related work on semi-supervised learning in the paper (see [4] for examples). It would be better to compare with Manifold Mixup [1], UAT [4] in the experiments.


Minor:

Page 5, " further break the correlation between $\delta$ and $y'$", what is $\delta$ here? I did not find the definition above the sentence. The notation is directly used without any explanation in advance.


References
[1] Manifold Mixup: Better Representations by Interpolating Hidden States, ICML 2019
[2] MixUp as Directional Adversarial Training
[3] On Adversarial Mixup Resynthesis, NeurIPS 2019
[4] Are Labels Required for Improving Adversarial Robustness?, NeurIPS 2019


**Experience Assessment:**

I have read many papers in this area.

**Review Assessment: Checking Correctness Of Derivations And Theory:**

I carefully checked the derivations and theory.

**Review Assessment: Checking Correctness Of Experiments:**

I carefully checked the experiments.

**Review Assessment: Thoroughness In Paper Reading:**

I read the paper thoroughly.

---

> ### Author Response · Authors · 2019-11-08
> **Thank you for the comments! We have addressed them in the sequel and in the revised paper**
>
> A1: Thanks for the reference and we have cited it and added discussions (Section 3.4) and comparisons (Section A.10) to our revised paper.
> Firstly, as you have noticed, the adversarial label interpolation is one difference with Manifold Mixup.
> Secondly, even when focusing on $x$, although related, the proposed approach is still different from Manifold Mixup from the following aspects. While both methods have an interpolation step in feature space, they have different motivations and are used differently. Manifold Mixup performs feature interpolation (mixup) and the mixup feature is passed to the subsequent layers till the cross-entropy loss. The cross-entropy loss-induced gradients backpropagate through the interpolation for model training. Manifold Mixup plays the role of a regularizer in order to obtain neural networks with smoother decision boundaries at multiple levels of representation.
> In terms of adversarial robustness, "Manifold Mixup did not significantly improve robustness against stronger, multi-step attacks such as PGD" (quoted from the Manifold Mixup paper, page 8), and was \emph{combined} with adversarial training by them in [5] to mitigate this issue.
>
> Differently, the proposed method uses features interpolation for \emph{inducing input perturbation}, which is done by backpropogating the gradient from feature space distance Eqn.(8) to the input (Figure 1). The perturbed inputs are then used as new inputs in place of the original ones for model training. Furthermore, empirically different from Mixup and Manifold-mixup, the proposed adversarial image interpolation scheme clearly helps to improve model robustness as shown in Table 5 and results below.
>
> A2: We propose a simple and practical approach that achieves state-of-the-art performance on defenses. We have provided an interpretation of the proposed method in Section 3.3 and Appendix A.5 from the perspective of robust and non-robust feature analysis of Madry et al. (NeurIPS 2019).
> The effectiveness can be attributed to the following two aspects:
> 1) the proposed method not only reduces the correlations between the perturbation $\delta$ and label $y$ as in conventional PGD adversarial training, but also reduces the correlations between $\delta$ and target label $y'$, thus leveraging two complementary aspects for improving model robustness
> 2) the proposed approach uses an adversarial interpolation scheme which is not directly connected with the classification loss as in conventional PGD adversarial training, thus mitigating the problem of label leaking as suffered by the conventional PGD-based training.
> We do agree with the reviewer that that a nice and more rigorous theory is useful for further understanding and improving of the proposed method and we will work on it as our next step.
>
> A3: Thanks for the references.  We have added discussion and comparisons in our revised paper (Section 3.4, Section A.10).
>
> ***Discussion***
>
> [2] is another submission to ICLR20 ( https://openreview.net/forum?id=SkgjKR4YwH ).
> It presents an interpretation of MixUp as belonging to a class highly analogous to adversarial training. Without evaluation of adversarial robustness provided in the paper, [2] is mainly on interpreting and understanding of Mixup, which is different from our goal of improving model robustness.
>
> [3] is on image generation, where the purpose of feature interpolation in [3] is to introduce human sensible visual attributes in the generated image, and the image is generated by forwarding the mixed feature through a decoder in the auto-encoding framework.
> Differently, the purpose of feature interpolation in the proposed method is to introduce human imperceptible, non-robust features in the generated image, which is generated by back-propagating the feature difference through the feature extractor (c.f. Figure 1).
>
> [4] is an interesting work that explores an orthogonal direction of improving adversarial generalization by using additional unlabeled data, which could potentially be used together with our approach.
>
> ***Comparison ***
>
> Attacks  ||  Mixup | Manifold-mixup  | IAT-mixup [5] | IAT-manifold-mixup [5] | UAT   | Adv-Interp
> ———————————————————————————————————————————————
> Natural  ||    96.8   |	      96.9               |        93.6          |               93.5                     | 85.9 . |   90.3
> FGSM     ||    67.4   |	      61.6               |        66.2          |               64.8                     |    -	   |   78.0
> PGD20   ||     0.7    |	        1.7               |        50.1          |               44.8 		      | 62.2  |   73.5
>
>
> A4: Thanks for the comments.
> $\delta$ denotes the perturbation added to the image $x$. It was in the caption of Table 7. We have added explanation to $\delta$ in the revised version of the paper.
>
> References
> [5] Interpolated Adversarial Training: Achieving Robust Neural Networks without Sacrificing Too Much Accuracy arXiv:1906.06784 2019

---

### Public Comment · ~Anthony_Wittmer1 · 2019-10-03
**Nice work and obfuscated gradients or not?**



It's an interesting work, the results are impressive. However I have some confusion about the experimental results.

1. From the results in Table 2, Adv-Interp (training with the attack epsilon 8) shows great robustness on PGD attack with epsilon 20. In my opinion, it is unconvincing. Could the authors explain what the reason for the robustness is? How does the Adv-Interp model obtain great robustness in the perturbation bounded, which is larger than that of the training process. The only reason I can think of is the obfuscated gradients, where the model mask the gradient on which most attacks rely [1]. In addtion, how about increasing the epsilon from 8 into 255 in the evaluation? I think the Adv-Interp model still shows greate robustness.

2. To break the model based on the obfuscated gradients, it is a better choice to evaluate on the gradient-free attack, such as Nattack [2]. But based on the observation of Section 4.4, the Adv-Interp model shows great robustness on black attacks. Have the authors tried Nattack[2] to perform the evaluation for black attacks, which break most defense methods with nearly 100% attack success rate.

[1] Obfuscated Gradients Give a False Sense of Security: Circumventing Defenses to Adversarial Examples. ICML 2018
[2] NATTACK: Learning the Distributions of Adversarial Examples for an Improved Black-Box Attack on Deep Neural Networks. ICML 2019

---

> ### Author Response · Authors · 2019-10-05
> **NOT obfuscated gradients-based false sense of robustness**
>
> Thanks for your comments. As we have already verified through extensive experiments (Table 1, 2 and Section 4.4) that the improved performance is due to improved robustness, and not due to obfuscated gradients.
>
> 1. While it is natural to expect the performance (accuracy) will decrease with an increasing attack epsilon, the decreasing speed could be different for different models. In general, a good robust model should not only perform well under the setting that is similar to training, but can also generalize reasonably well beyond it.
>
> Figure 1 from [1] demonstrated the accuracy-epsilon curves for epsilon <=8 (8/255=0.031) as a way to analyze and compare different methods.
> Table 2 from our paper actually further “extends” this analysis to the range of epsilon <=20, and the results in Table 2 actually demonstrated the effectiveness of the proposed approach in terms of generalization beyond the training epsilon.
> If we further extend the range to epsilon <=255 (as you suggested), the accuracy of all methods will eventually drops to zero. The difference is that the proposed model approaches it in a more graceful manner, which is a manifestation of improved robustness.
>
> 2. We have already evaluated against four representative types of black-box attacks following previous works, including two gradient-based and two gradient-free methods. These results together with the white-box evaluations already clearly verified the improvement is due to improved robustness instead of obfuscated gradients. Nevertheless, we evaluated the performance of the proposed Adv-Interp model against the Nattack method [2] over 1000 test images (due to its computational cost), and the accuracy of our model is 76%. This result together with the white-box results from Table 1 reconfirm the inherent improved model robustness, instead of a false sense of robustness due to obfuscated gradients.

---

### Public Comment · ~Zejia_Weng1 · 2019-11-03
**Interesting Work but I met some trouble reproducing the result**

Very impressive results!

I tried to implement your proposed method and did experiments on cifar10. But the robust accuracy is 0.5346(wideResnet18) , which may be caused by some mistakes in my code or my high-parameters setting. My code is here "https://github.com/wengzejia1/Interpol".
It would be great if you can spare some time to check it. It would be better if you could release a more detailed full code!

Best.

---

> ### Author Response · Authors · 2019-11-04
> **Thanks for your interest in our work! And please make sure to follow the code provided in our paper for your convenience**
>
> The code we provided in Appendix A.1 has already included all the details needed for implementing the proposed method, due to its inherent simplicity.
>
> Upon initial checking of your implementation, it seems that you didn’t follow the code we provided.
> These differences could be easily spotted by comparing your code with the one we provided.
>
> The first difference is the feature extractor module. As shown in Line 20 of our code, ‘feature’ mode is used for feature extraction, which “is implemented by excluding the last layer for the logits” as described in the implementation details (beginning of the second paragraph of Section 4). Also, the step size 'epsilon' in Line 24 of our code is missing in your implementation. These should be the major bugs in your implementation and please correct them.
> Other minor differences include the base-net (we used WRN-28-10), initialization (we used uniform as in Line 16 of our code), learning rate schedule etc. While these should be less impactful, it might be easier for you to first use the same setting to make sure your code is bug-free before experimenting with different settings.
>
> You should be able to get similar results after fixing the discrepancies (by following our code) as mentioned above.

---

### Public Comment · ~Daquan_Lin1 · 2019-11-08
**I created a new attacker, it can beat Adv_inter with robust accuracy remain 38%.**

**Idea:
The idea is very simple, and how to generate adversarial images is same with your method.
Suppose a clean image X, and X's label is Y use random start we can get X+R, where R is uniform noise range in [-eps, eps].
Suppose a target clean image X_t, X_t's label is Y_t, which not equal to Y.

X_adv = X+R
feat_target = net(X_t, 'GAP')   // feat_target is the output of global average pooling(GAP) in last Conv. layer
for i in num_step:
     feat = net(X_adv, 'GAP')
     loss = cos_dist(feat, feat_target)
     loss backward
     X_adv = X_adv - step_size * sign(grad_X_adv)
     X_adv clip with [X-eps, X+eps]
     X_adv clip with [0, 1]  # or other e.g. [-1, 1] as your setting

**Experiment:
On CIFAR10,
Each clean image will have 100 random target images to run attack,
step_size=1/255, eps=8/255, num_step=100

**Codes:
https://github.com/Line290/FeatureAttack

**Results:
| Attack type             | clean | FGSM | PGD20 | CW20 | FeatureAttack100          |
|-----------------            |-------  |--------  |-----------|---------|---------------------------------|
| Feature Scatter[1] | 90.3   | 78.4    | 71.1      | 62.4   | 37.4(5000 test images)  |
| Adv_inter                | 90.5   | 78.1    | 74.4       | 69.5   | 38(5600 test images)    |
| Madry                     | 87.25 |             | 45.87    |            | 46.37                                |

**Discussion:
I checked Feature Scatter's model parameters, Madry's and vanilla training's, and found L1(Feat_Scat) < L1(Madry) << L1(vanilla) w.r.t weights of FC layer, which size is 10*640 in wideResnet18. I guess different adversarial training methods will cause different intensities of gradient vanish in last several layers.

**Reference:
[1]Haichao Zhang Jianyu Wang, Defense Against Adversarial Attacks Using Feature Scattering-based Adversarial Training, NeurIPS 2019
[2]Madry, et al. Towards Deep Learning Models Resistant to Adversarial Attacks, ICLR 2018

---

> ### Author Response · Authors · 2019-11-08
> **Thank you and could you add results on Madry model?**
>
> Thank you for showing another useful application of Adv-Interp on the attack side. Could you please add the results on Madry model to the Table so that readers will have a complete picture? Thanks!

---

> > ### Public Comment · ~Daquan_Lin1 · 2019-11-08
> > **I am doing experiment on Madry's model.**
> >
> > Please wait for several hours.

---

> > ### Public Comment · ~Daquan_Lin1 · 2019-11-08
> > **I added the results on Madry model to the Table.**
> >
> > Codes to attack Madry model is added to Repo: FeatureAttack

---

### Author Response · Authors · 2019-11-08
**Adversarial Interpolation Training: code and model released**

We would like to thank all the reviewers for their efforts in reviewing our paper and for recognizing the contribution of our work. We also thank fellow researchers who have shown interest in our work and have been interacting with us through openreview.net.

We have addressed the comments from the reviewers and other readers individually in the sequel.
We have also released our code and model to further contribute to the community and also in the spirit of Carlini, Nicholas, et al. "On evaluating adversarial robustness" 2019.

The code and model are released here: https://github.com/Adv-Interp/adv_interp

---

### Decision · Program_Chairs · 2019-12-19

**Decision:**

Reject

**Comment:**

Reviewers agree that the proposed method is interesting and achieves impressive results. Clarifications were needed in terms of motivating and situating the work. Thee rebuttal helped, but unfortunately not enough to push the paper above the threshold. We encourage the authors to further improve the presentation of their method and take into accounts the comments in future revisions.